# Radiomics Enabler®, an ETL (Extract-Transform-Load) for biomedical imaging in big-data projects

**Karine Seymour**
Medexprim, Toulouse, France
kseymour@medexprim.com
http://www.medexprim.com

**Pierre Payoux**
Département de médecine nucléaire,
Hôpital de Purpan,
Place du Docteur Baylac,
TSA 40031,
31059 Toulouse Cedex 9
payoux.p@chu-toulouse.fr

## Abstract

Clinical data warehouses are now routinely used by large hospitals. They allow researchers to pool medical data from millions of patients to create unified and harmonized patient cohorts for clinical trials and epidemiology studies. In biomedical imaging, although there is a need to extract large amounts of data to help develop decision tools based on artificial intelligence, the identification, extraction and anonymization of relevant sequences is traditionally very time-consuming, as it is done patient by patient. We developed Radiomics Enabler®to streamline this process. We compared the extraction and anonymization process with and without Radiomics Enabler®on 118 imaging exams. We found that the operator's time required using his PACS (Picture Archiving and Communication System) workstation was 5 hours and 37 minutes. The time was reduced to only 17 minutes with Radiomics Enabler®. Radiomics Enabler®is available as open-source and can be integrated with any clinical data warehouse.

## 1 Introduction

Existing biomedical imaging exams stored in hospitals' clinical PACS[1] are treasures of information for research. In particular, exploiting exiting imaging data is useful for radiomics studies and to develop imaging cognitive tools.

Radiomics consists of extracting large amounts of quantitative descriptors out of biomedical imaging exams using image post-processing pipelines and finding correlations between these descriptors and associated data (genotypes and other phenotypes) (Lambin et al. 2002). It is a quickly evolving field as illustrated by the evolution of number of publications shown in Figure 1.

At the same time, a number of academic research teams and private companies are developing cognitive imaging tools which can "learn as they go", using deep learning technologies[2,3]. These deep-learning algorithms need to be initiated with massive amounts of routine biomedical images and associated data representative of patient and disease variability.

Major hospitals are starting to get equipped with clinical data warehouses, enabling them to pool medical data from millions of patients to create unified and harmonized patient cohorts for clinical

---

[1]PACS: Picture Archiving and Communication System
[2]https://www.forbes.com/sites/bernardmarr/2017/01/20/first-fda-approval-for-clinical-cloud-based-deep-le
[3]http://www.icadmed.com/

1st Conference on Medical Imaging with Deep Learning (MIDL 2018), Amsterdam, The Netherlands.

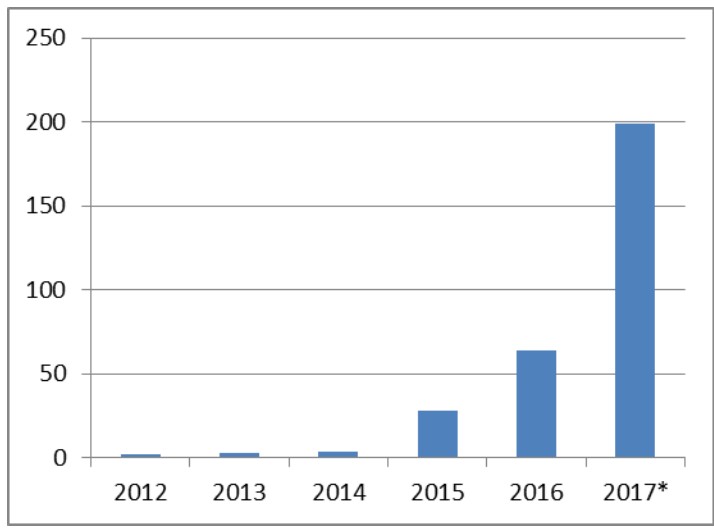

**Figure 1:** Evolution of number of publications on radiomics recorded in Pubmed.

trials and epidemiology studies. Integrating biomedical imaging data, however, remains a challenge. Identifying and extracting relevant sequences from the hospital's PACS, then de-identifying them and loading them on a post-processing pipeline is traditionally very time-consuming as it is done patient by patient.

We developed Radiomics Enabler®, an ETL (Extract-Transform-Load) for medical imaging, which streamlines this process. Radiomics Enabler®is made freely available to the imaging community under GNU Affero General Public License v3[4]. The source code is available on https://bitbucket.org/medexprim/radiomics-enabler.

The objective of this study is to qualify and quantify the impact of Radiomics Enabler®by comparing the extraction and de-identification process with and without the solution.

## 2   Material and methods

Radiomics Enabler®is a web application which Medexprim developed in collaboration with researchers and clinicians from the Toulouse University Hospital. It allows a user to perform a multi-criteria search on the PACS, filter the results and select the relevant sequences. The extraction, de-identification and secured routing of selected images is then entirely automatized. We also developed a standard API (Application Programming Interface), so that Radiomics Enabler®can be integrated within a Clinical Data Warehouse (CDW) to create unified and harmonized patient cohorts, with a complete access to their clinical and imaging data.

The application was developed in python, using the Django[5] and Celery[6] frameworks. It uses the dcmtk DICOM open-source library[7]. The data and user interface layers are separated (Model View Controller architecture), so that the core software could easily be integrated within another application. It is also possible to deploy the layers on separate servers to allow for different levels of network access. This would be particularly useful in a regional PACS setting where you need to provide user web access, while ensuring that data are securely stored.

The application is functionally structured around research projects, with user rights attached to each project as illustrated in Figure 2 . For each project, users constitute lists of DICOM studies and/or series which are to be batch extracted from the PACS. Each batch has a status associated to it: in preparation, processing, interrupted, complete, in error or archived.

---

[4]https://www.gnu.org/licenses/agpl-3.0.en.html
[5]https://www.djangoproject.com
[6]http://www.celeryproject.org/
[7]http://www.dcmtk.org/dcmtk.php.en

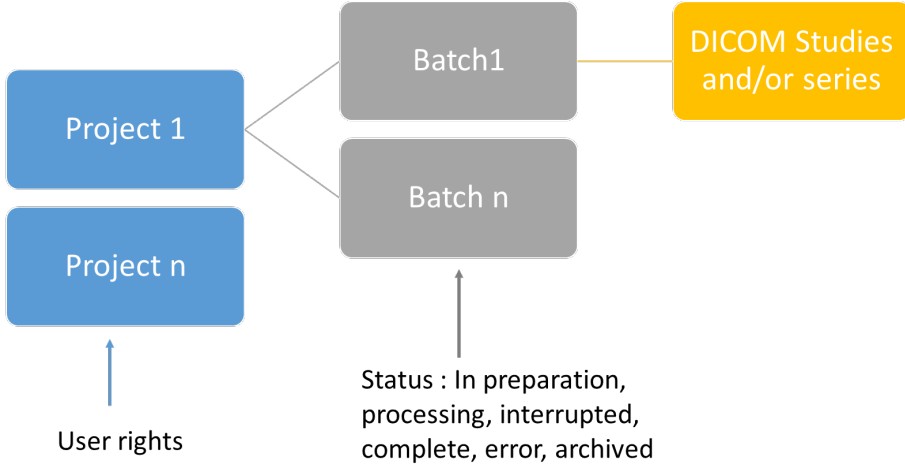

**Figure 2:** Structure of Radiomics Enabler®around projects and batches.

Figure 3 illustrates the process of extracting, de-identifying and processing medical imaging using Radiomics Enabler®.

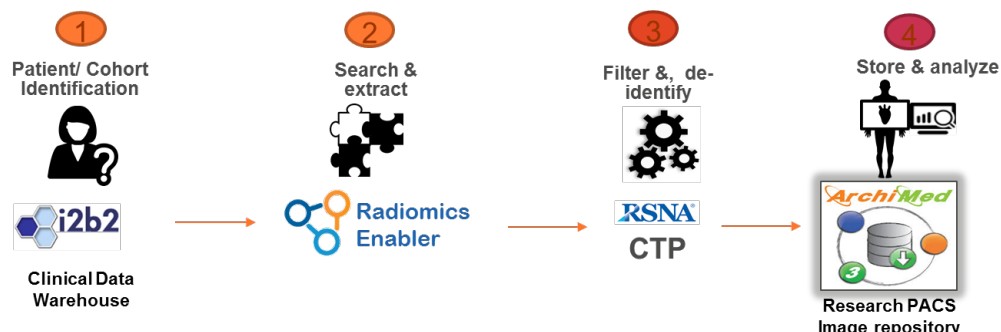

**Figure 3:** The process of extracting, anonymizing and processing medical imaging exams for secondary use in research.

The typical stages are:

1. The user identifies a list of patients that is relevant for his study by searching through clinical reports, biological analysis results, diagnosis codes, etc. This typically is done using a CDW such as i2b2 (Murphy and al.2010) or ConSoRe (Heudel and al. 2016);

2. The user calls Radiomics Enabler®and adds additional search criteria, such as modality or exam type, to the list of patients, as illustrated in Figure 4, then queries the PACS (DICOM[8] c-find transaction). He can filter the results based on the DICOM fields "study description" and/or "series description", then adds the relevant series to a "batch". He can add more exams or specific images sequences to his batch until he decides the batch is ready to be extracted from the PACS.

3. The extraction (DICOM c-move), then de-identification and routing to the post-processing pipeline according to a pre-defined set of rules is entirely automatic. The de-identification part is done with the open-source solution Clinical Trials Processor (CTP) from the Radiology Society of North America (RSNA)[9].

4. Medical images are then ready to be analysed on any post-processing pipeline.

---

[8]DICOM (Digital Imaging Communication in Medicine) standard :`http://dicom.nema.org/standard.html`

[9]`https://www.rsna.org/ctp.aspx`

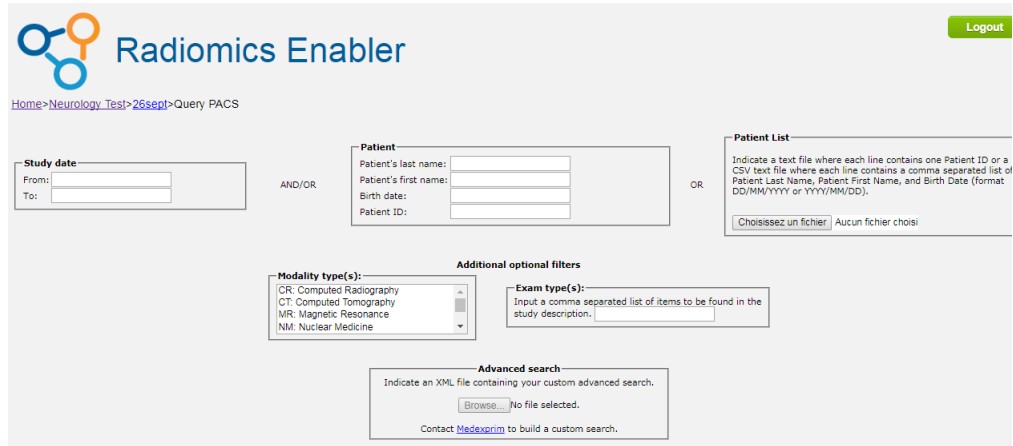

**Figure 4:** A screenshot of a typical query dialog in Radiomics Enabler®.

At any time, a user can access his research projects (user rights are defined per research project) and see the status of his extraction batches as illustrated in Figure 5 . A click on the batch displays the list of exams or sequences that are part of the batch.

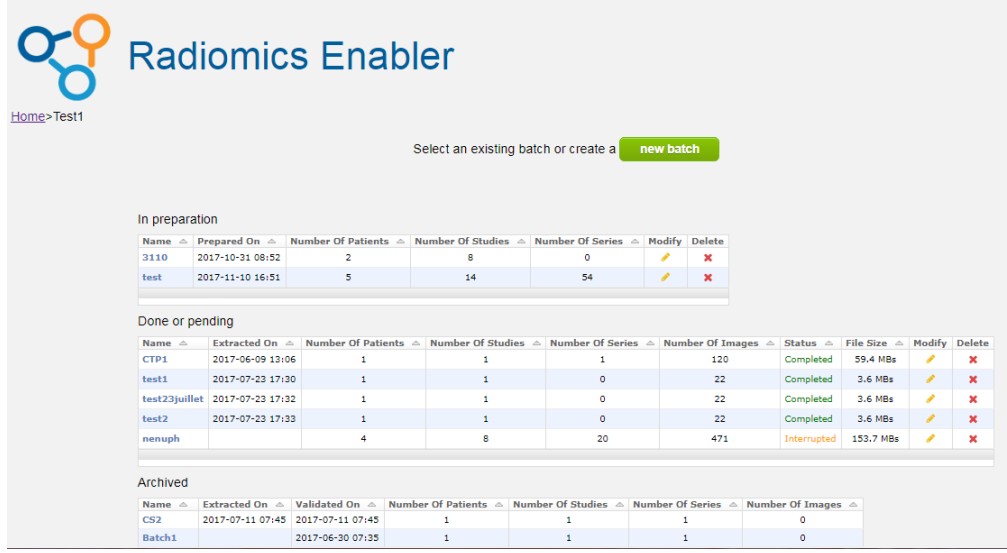

**Figure 5:** A screenshot of the summary page of a project showing the associated batchs in Radiomics Enabler®.

In a recent study on epilepsy, we compared the operator's time to extract, de-identify and export on an external medium 118 imaging exams (PET-CT and MRI exams) from a list of known patients:

1. using the tools provided by our PACS workstation,
2. using Radiomics Enabler®.

For the first method, we measured the time required to extract 12 exams and extrapolated the results for the 118 exams.

## 3   Results

Results are reported in Table 1.

**Table 1:** Comparison of methods in extracting and de-identifying imaging data

|  | With the PACS workstation | With Radiomics Enabler® |
|---|---|---|
| Operator dependent time required for extracting, anonymizing and saving on an external medium | 5 hours and 37 min (extrapolation from 12 exams) | 17 min (2 min to create the list of patients, 15 min to select the right sequences) |
| PACS' workstation availability during the extraction process | Unavailable for its primary purpose (reading imaging exams) | Available |
| Traceability | Low | High |
| Risk of errors | High | Low |

When using our PACS workstation, each exam had to be opened and copied to an external medium. The export function includes the ability to modify the patient information from the DICOM header (de-identification). We measured the time required for 12 exams and extrapolated the results for the 118 exams. We calculated that it would have taken over 5 1/2 hours of continuous operator's time. Moreover, when used to extract data, our workstation is not available for primary reading. At last, using this method requires to keep a side record of each extraction we do, which is prone to errors.

With Radiomics Enabler®, the only operator dependent action is the selection of the relevant sequences. Integrating the list of patients and querying the PACS was very fast (2 min). For our study, we needed to extract only MRI scans that had been done within a year of the PET-CT scan. We did this selection manually from the list of exams that was displayed on Radiomics Enabler®. This is what took the longest in the process (15 minutes). The actual extraction and de-identification process, which took about 2 hours, did not have any user impact, since it is automated. In addition, everything is traced in Radiomics Enabler®

## 4    Conclusion and perspective

Radiomics Enabler®has proved to be a very useful tool to streamline the process of extracting and de-identifying biomedical imaging data in retrospective big-data projects. The more data that is extracted, the greater the impact. Freed from time-consuming low value-added tasks, researchers can now focus on their core expertise: data analysis and medical discoveries.

We recently released the software as open-source, hoping to foster a large community of users and contributors. We also demonstrated our ability to integrate with different clinical data warehouses such as i2b2 (Murphy and al.2010) and ConSoRe (Heudel and al. 2016) to unleash the potential of medical images combined with clinical data.

There exists federations of clinical data warehouses like SHRINE (Weber and al. 2009). Similarly, we are looking to develop a collaborative platform federating individual instances of Radiomics Enabler®to better serve the requirements of global multi-centre research projects.

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
