# OpenReview forum: "Radiomics Enabler®, an ETL (Extract-Transform-Load) for biomedical imaging in big-data projects"
_MIDL.amsterdam/2018/Conference — Submitted to MIDL 2018_

### Review · AnonReviewer2 · 2018-05-09
**no deep learning**

**Rating:** 1
**Confidence:** 3

**Review:**

The paper describes an open source tool, “Radiomics Enabler”, which makes it easier for researchers  to perform a multi-criteria search on the PACS, filter the results and select the relevant image studies, anonymize the images and save them to an external medium. There is also a small experiment that estimates the time gain – from over 5 hours with standard PACS tools to 17 minutes for a study of 118 exams. Though this tool can certainly be useful for medical imaging researchers, it is not specific to deep learning nor uses any deep learning, and I therefore believe it is out of scope for this conference (From the website: “Full papers in the conference track contain methodological developments or well-validated applications of deep learning algorithms in medical imaging”).

**Special Issue:**

No

---

### Review · AnonReviewer1 · 2018-05-09
**Interesting tool, but not a fit for MIDL**

**Rating:** 1
**Confidence:** 3

**Review:**

This work presents Radiomics Enabler, a platform that allows researchers to quickly extract data from PACS given study criteria. This is an improvement on the current practice of a researcher manually searching PACS and downloading studies individually.

A small experiment was conducted that involved the extraction of 12 cases - results were reported on 118, but this was just an extrapolation. Why not test on a larger study? Also, the abstract shouldn't mention 118 cases, as this is misleading.

Overall, this is likely a very useful tool, but this work isn't a good fit for MIDL, as there is no connection to deep learning.

Other notes:
- Page 1 footnotes require formatting
- Figure 3 has a caption above and below the figure

**Special Issue:**

No

---

### Review · AnonReviewer3 · 2018-05-10
**Tool for PACS data retrieval. Out of scope for MIDL**

**Rating:** 1
**Confidence:** 2

**Review:**

This paper describes a tool to efficiently extract and anonymize data from a hospital PACS system. The tool allows researchers to search in a clinical PACS system for data (e.g. for deep learning research) and efficiently extract and store this data. It is reported that using this tool saves time compared to manually saving and anonymizing each data set.
The method is very specific for the use case where researchers have access to a clinical PACS system with PHI and need to anonymize data. In my understanding this is often not the case. Most research institutions have already systems in place for this and for situations where this is not the case, it will most often not be the medical imaging deep learning researchers that are collecting the data. In those cases, data will be collected and anonymized by a clinical collaborator. Because of that I believe this paper is not of interest to the audience of the MIDL conference. Also, the tool is evaluated in only one hospital and in a study where only 12 cases are retrieved. When presenting this, I would suggest to first do a larger study across multiple hospitals, PACS systems, and research situations to demonstrate the value of this tool for the community.


**Special Issue:**

No

---

### Decision · Program_Chairs · 2018-05-15
**Paper24 Acceptance Decision**

Reject